

# 16S rRNA gene sequencing reveals effects of photoperiod on cecal microbiota of broiler roosters

Jun Wang[*], Lucky T. Nesengani[*], Yongsheng Gong, Yujiang Yang and Wenfa Lu

College of Animal Science and Technology, Jilin Agricultural University, Changchun, China
[*] These authors contributed equally to this work.

## ABSTRACT

Photoperiod is an important factor in stimulating broiler performance in commercial poultry practice. However, the mechanism by which photoperiod affects the performance of broiler chickens has not been adequately explored. The current study evaluated the effects of three different photoperiod regimes (short day (LD) = 8 h light, control (CTR) = 12.5 h light, and long day (SD) = 16 h light) on the cecal microbiota of broiler roosters by sequencing bacterial 16S rRNA genes. At the phylum level, the dominant bacteria were *Firmicutes* (CTR: 68%, SD: 69%, LD: 67%) and *Bacteroidetes* (CTR: 25%, SD: 26%, and LD: 28%). There was a greater abundance of *Proteobacteria* ($p < 0.01$) and *Cyanobacteria* ($p < 0.05$) in chickens in the LD group than in those in the CTR group. A significantly greater abundance of *Actinobacteria* was observed in CTR chickens than in SD and LD chickens ($p < 0.01$). The abundance of *Deferribacteres* was significantly higher in LD chickens than in SD chickens ($p < 0.01$). *Fusobacteria* and *Proteobacteria* were more abundant in SD chickens than in CTR and LD chickens. The predicted functional properties indicate that cellular processes may be influenced by photoperiod. Conversely, carbohydrate metabolism was enhanced in CTR chickens as compared to that in SD and LD chickens. The current results indicate that different photoperiod regimes may influence the abundance of specific bacterial populations and then contribute to differences in the functional properties of gut microbiota of broiler roosters.

## INTRODUCTION

Photoperiod is defined as the relative amount of light per day to which an organism is exposed (*Lee, Park & Lee, 2017*). This period of exposure to light can influence different aspects of physiology in avian as well as mammalian species, such as reproduction, behavior, and immune functions, to different magnitudes (*Pittendrigh & Daan, 1976*; *Walton, Weil & Nelson, 2011*). Following photoperiod, animals tend to undergo a suite of adaptive responses by altering their physiology and reproductive state for survival (*Walton, Weil & Nelson, 2011*). An increased photoperiod length has been reported to result in lower incidence of skeletal diseases and increase in weight gain with slower growth in broiler chickens (*Classen, Riddell & Robinson, 1991*). Decreasing or increasing photoperiod can also be used

Corresponding author
Wenfa Lu, wenfa2004@163.com

to reduce the early growth rate of broilers but allow them to compensate as they approach market age (*Downs et al., 2006*). Photoperiod was also found to affect the physiology of chickens; birds exposed to short days had a higher expression of gonadotropin-inhibitory hormone expression compared to birds exposed to long days (*Dixit, Singh & Byrsat, 2017*). Furthermore, it was reported that long photoperiods promote the development of the gonads in poultry (*Kang & Kuenzel, 2015*), although the exact mechanism underlying this effect is still unclear. Given the many roles that photoperiod plays in various aspects of the physiology of avian and mammalian species, it is of scientific interest to evaluate its role in other inadequately explored aspects, such as the bacterial structure and functional properties of the gut microbiota.

Gut bacteria, which form part of the gut microbiota, have been shown to play important roles in digestion, metabolism, and health in avian species (*Waite & Taylor, 2014*; *Waite & Taylor, 2015*). Gut microbiota have been widely reported to be affected by factors such as diet and age (*Waite & Taylor, 2014*; *Zhao et al., 2017*; *Zhu et al., 2017*). However, other factors that may affect the structure and functional properties of chicken gut microbiota, such as photoperiod, have yet to be evaluated. Photoperiod may play a significant role in determining most physiological functions by altering the gut microbiota. At present, there is a gap in the knowledge on the role of photoperiod in gut microbiota structure and function. The present study evaluated the effect of photoperiod on the abundance, diversity, and predicted functional properties of cecal microbiota in broiler roosters by sequencing the 16S rRNA gene.

## MATERIALS AND METHODS

Ethical approval for the present study was obtained from the Ethical Committee of the Jilin Agricultural University, China.

### Photoperiod treatments

One hundred and twenty AA+ Broilers (20 weeks of age, average weight: 2,806 g) were randomly divided into three groups ($n = 40$) and subjected to different photoperiodic regimes for five weeks. Group I was designated the Control group (CTR; 12.5 h Light:11.5 h Dark, i.e., lights on at 08:00 a.m. and lights off at 08:30 p.m.), Group II the Long-day photoperiod group (SD; 16 h Light:8 h Dark, i.e., lights on at 04:00 p.m. and lights off at 08:00 a.m.), and Group III the Short-day photoperiod group (LD; 8 h Light:16 h Dark, i.e., lights on at 08:00 a.m. and lights off at 04:00 p.m.). A 60 W incandescent lamp with an illuminating intensity of 30 lux was used as the source for artificial illumination and was positioned at the height of the head of standing birds. All the broiler roosters were maintained in cages of equal size. Each rooster was fed 115 g of commercial broiler diet per day for 20 weeks via restricted feeding before the experiment, and then the amount of feed was increased by 5 g every week. In order to ensure that each rooster was fed the same amount of diet, each rooster was kept in an individual cage. Water was provided ad libitum during the whole experimental period.

## Sample collection

All the roosters were slaughtered at about 25 weeks of age. Luminal cecum contents were collected from seven randomly selected broilers from each group. All samples were harvested within 30 min after slaughter and immediately frozen in liquid nitrogen. The frozen luminal samples were stored in a freezer at −80 °C; until further use.

## DNA extraction and 16S rRNA amplification

Samples were allowed to thaw at room temperature before DNA extraction. Total genomic DNA was extracted using the Fast DNA SPIN extraction kits (MP Biomedicals, Santa Ana, CA, USA), following the manufacturer's instructions. DNA concentration was evaluated by measuring optical density using Nano-Drop 2000 (Thermo Electron Corporation, Waltham, MA, USA) at wavelengths of 260 and 280 nm. The integrity of the DNA extracts was assessed by electrophoresis on 1.0% agarose gels. The V4–V5 regions of the bacterial 16S rRNA gene were amplified from the total microbial genomic DNA via PCR using the forward primer 515F (5′-GTGCCAGCMGCCGCGGTAA-3′) and the reverse primer 907R (5′-CCGTCAATTCMTTTRAGTTT-3′) . The amplification was carried out in 25 μL reactions containing 5 μL of Q5 reaction buffer (5×), 5 μL of Q5 High-Fidelity GC buffer (5×), 0.25 μL of Q5 High-Fidelity DNA Polymerase (5 U/ μL), 2 μL of dNTPs (2.5 mM), 1 μL each of the forward and reverse primer (10 uM), 2 μL of DNA template, and 8.75 μL of ddH$_2$O. PCR conditions were as follows: initial denaturation at 98 °C for 2 min; followed by 25 cycles of denaturation at 98 °C for 15 s, annealing at 55 °C for 30 s, and extension at 72 °C for 30 s; and then a final extension at 72 °C for 5 min. The PCR products were separated on 2% agarose gels and subsequently extracted from the gels. Samples with a bright band with a size between 200–450 bp were chosen for downstream experiments. PCR products were purified using a GeneJET Gel Extraction Kit (Thermo Scientific, Waltham, MA, USA). Products were quantified using a PicoGreen dsDNA Assay Kit (Invitrogen, Carlsbad, CA, USA). After quantification, the amplicons were pooled in equal amounts, and pair-end 2 × 300-bp sequencing was performed using the Illlumina MiSeq platform and a MiSeq Reagent Kit v3 at Shanghai Biotechnology Co., Ltd (Shanghai, China).

## Bioinformatics and statistical analysis

The quality control and analysis of the sequences were performed using the software Quantitative Insights into Microbial Ecology (QIIME, v1.8.0) (*Caporaso et al., 2010*). The paired-end reads from the DNA fragments were merged using FLASH (*Magoc & Salzberg, 2011*). The UCLUST (*Edgar, 2010*) clustering method was used to cluster operational taxonomic units (OTUs) with ≥97% sequence identity. OTU classification was conducted by running a BLAST search against the Greengenes Database (*DeSantis et al., 2006*) using the representative sequence set as a query (*Altschul et al., 1997*). To minimize the difference in sequencing depth across samples, an averaged, rounded, and rarefied OTU table was generated by averaging 100 evenly re-sampled OTU subsets under the 90% of the minimum sequencing depth. These were then used for further analysis.

Bioinformatics and statistical analyses were performed using the QIIME and R packages (v3.2.0). The alpha-diversity indices (Chao1, ACE metric, Shannon diversity

**Table 1** The average alpha-diversity indexes (chao1, Simpson and Shannon index) of the data distribution.

| Group | Chao1 | | Simpson | | Shannon | |
|---|---|---|---|---|---|---|
| | Mean | STD | Mean | STD | Mean | STD |
| CTR | 1,799.593[*] | 256.3406 | 0.972857 | 0.00488 | 7.411429 | 0.281569 |
| SD | 1,461.779[*] | 310.5823 | 0.977143 | 0.00488 | 7.525714 | 0.227146 |
| LD | 1,729.097 | 224.2392 | 0.975714 | 0.007868 | 7.607143 | 0.223958 |

**Notes.**
[*]Numbers with asterisks are significantly different ($p$ value $< 0.05$).

index, and Simpson index) were calculated using the QIIME software to establish the abundance and diversity of the sequences. Beta-diversity was determined using unweighted UniFrac distance metrics to evaluate the structure and distribution of the microbial genetic communities among the samples (*Lozupone & Knight, 2005*; *Lozupone et al., 2007*). Differences in the Unifrac distances for pairwise comparisons among groups were calculated using Student's $t$-test and the Monte Carlo permutation test with 1,000 permutations. Significance was assigned when $p < 0.05$ and $p < 0.01$. The differences and similarities between the compared groups were evaluated using ANOSIM (analysis of similarities) in the R package "vegan" (*Oksanen et al., 2017*). A Venn diagram was generated using the R package "VennDiagram" to visualize the shared and unique OTUs among samples or groups. Functional genes were predicted using PICRUSt (phylogenetic investigation of communities by reconstruction of unobserved states) using high-quality sequences as the input (*Langille et al., 2013*).

# RESULTS

## Sequencing overview

A total of 21 samples were obtained from three groups ($n = 7$ per group) of broiler roosters and subsequently sequenced to generate V4–V5 16S rRNA gene profiles. A total of 398,445, 328,235, and 375,402 sequences were obtained for the CTR, SD, and LD groups, respectively. There was an average of 56,920, 46,890, and 53,628 reads per sample in the CTR, SD, and LD groups, respectively.

## Validation and structure determination of the sequences

The variation in data distribution between the groups was analyzed using ANOSIM, which indicated a significant difference ($p < 0.01$) between the three groups under unweighted Unifrac. The alpha-diversity indices (chao1, Simpson and Shannon index) are reported in Table 1, there was a significant difference between CTR and SD groups when comparing a chao1 indices mean. The results of the beta diversity analysis and the PLS-discriminant analysis are shown in Figs. 1 and 2, respectively. Samples from LD and SD indicated to be clustered similar where the CTR samples were different in NMDS analysis. The PLS-discriminant analysis indicated that the two groups are different with the exception of one sample from LD which was clustered with SD.

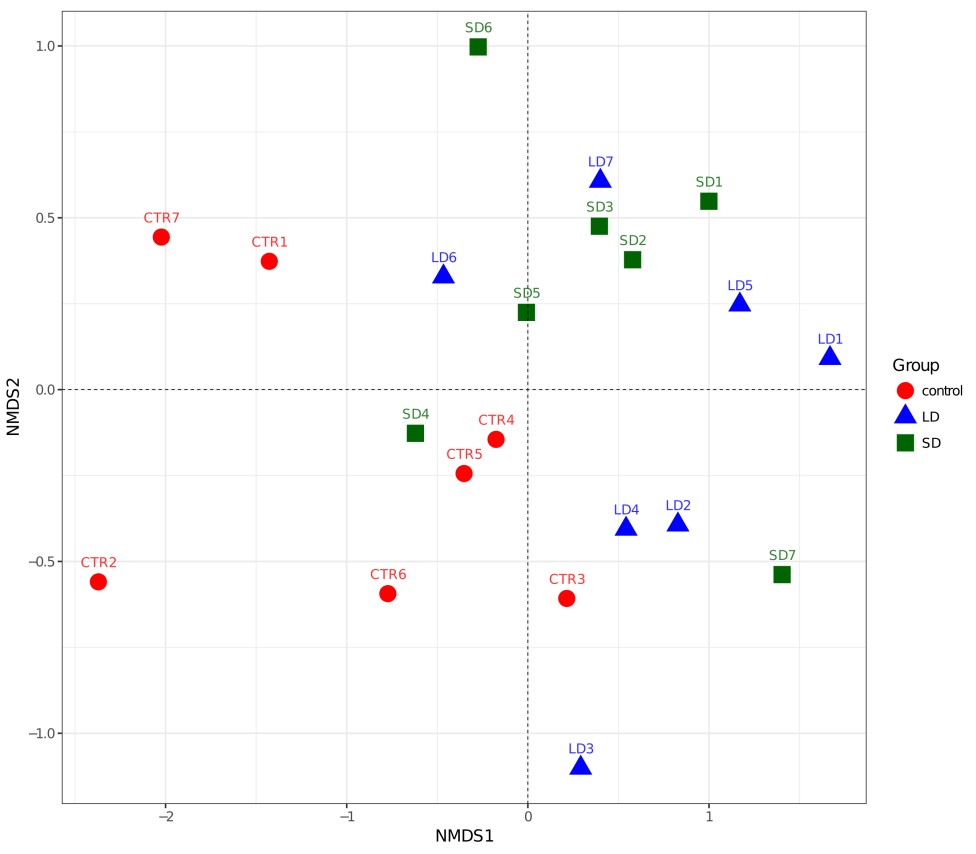

**Figure 1** **The beta diversity results of NMDS indicating the data distribution between the groups.** LD refers to the 8 h light group samples, CTR refers to the 12.5 h light and SD refers to the 16 h light group samples respectively.

## Abundance and significant difference between the three groups at the phylum level

The most abundant bacteria at the phylum level were *Firmicutes*, with abundances of 68%, 69%, and 67% in the CTR, SD, and LD groups, respectively, followed by *Bacteroidetes* with abundances at 25%, 26%, and 28% in the CTR, SD, and LD groups, respectively (Fig. 3). The other bacterial phyla had abundances lower than 3% in all groups at varying magnitudes. As shown in Fig. 2, *Proteobacteria* ($p < 0.01$) and *Cyanobacteria* ($p < 0.05$) were more abundant in LD chickens than in CTR chickens, while *Actinobacteria* was more abundant in chicken from the CTR group than in those from the LD group ($p < 0.01$). Between the CTR and SD groups, there was a significant difference in the abundance of *Actinobacteria*, which was more abundant in the CTR group than in the SD group ($p < 0.01$). *Deferribacteres* was more abundant in LD roosters than in SD roosters ($p < 0.05$). *Fusobacteria* and *Proteobacteria* were significantly more abundant ($p < 0.01$) in chickens from the SD group than in those from the CTR and LD groups, as indicated in Fig. 4.

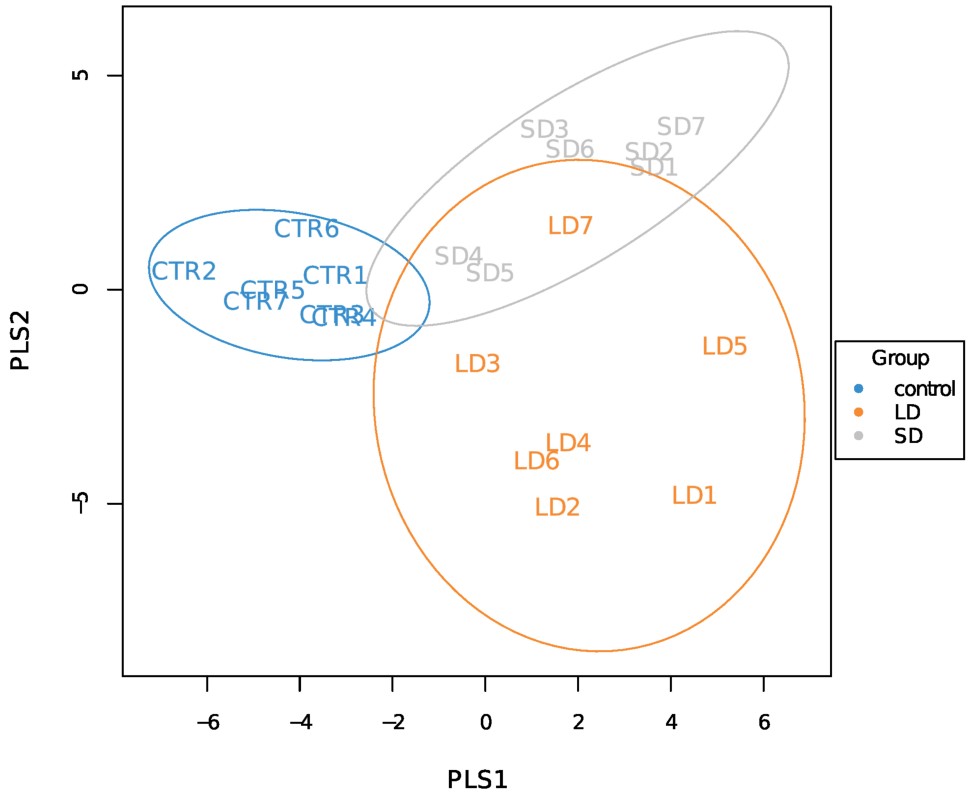

**Figure 2** **The PLS-discriminant analysis.** LD refers to the 8 h light group samples, CTR refers to the 12.5 h light and SD refers to the 16 h light group samples respectively.

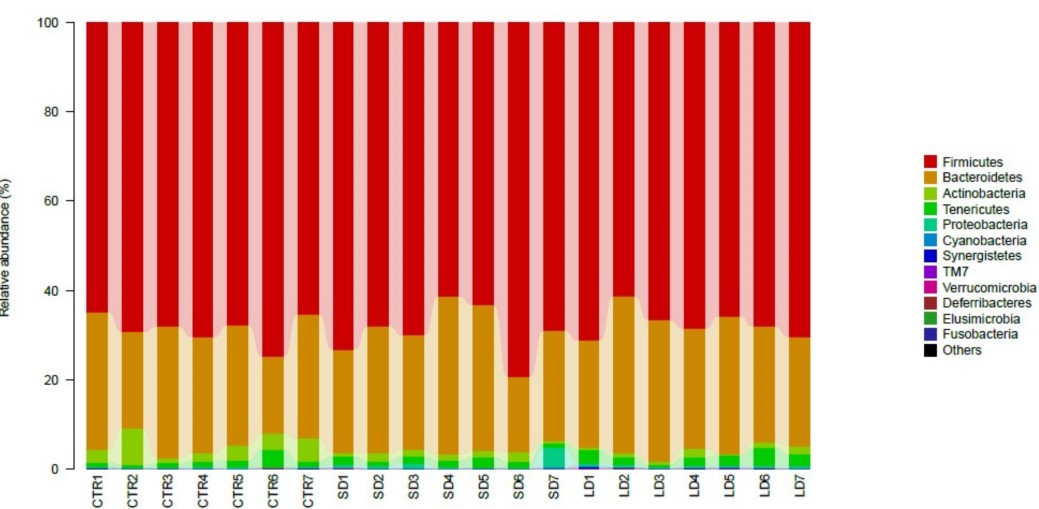

**Figure 3** **Taxonomic profiles of the microbial communities at the phylum level.** LD refers to the 8 h light group, CTR refers to the 12.5 h light and SD refers to the 16 h light group. Samples are presented along with the horizontal axis and relative abundance at the vertical axis.

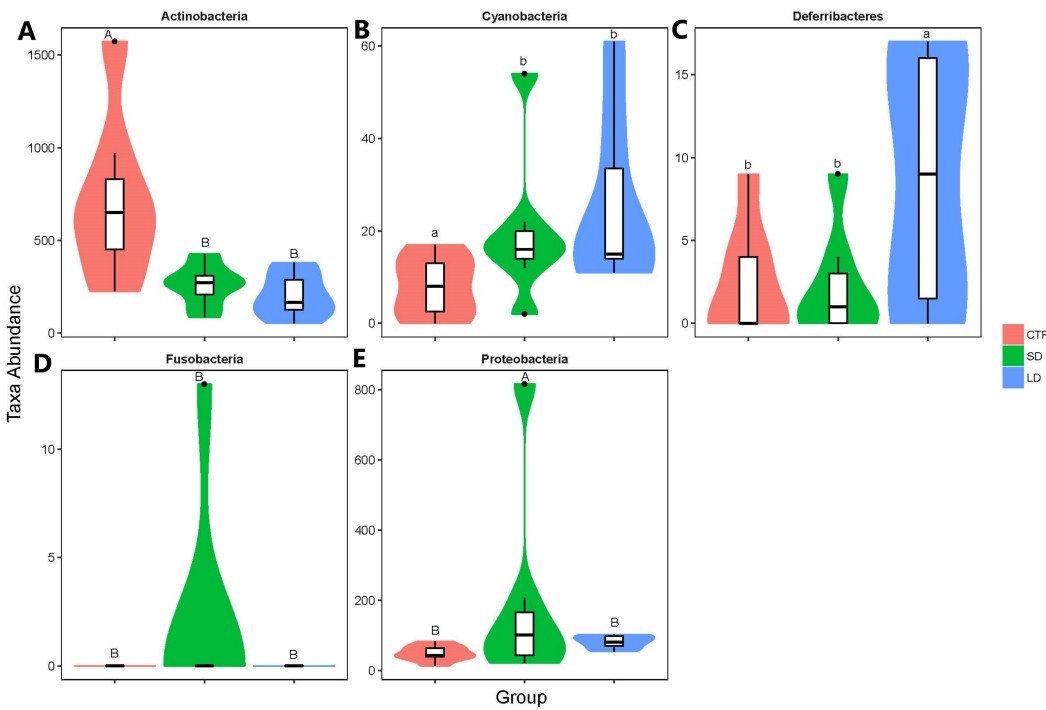

**Figure 4** **Taxonomic profiles of the notable significant different bacterium at the phylum level.**
(A) Actinobacteria, (B) Cyanobacteria, (C) Deferribacteres, (D) Fusobacteria, (E) Proteobacteria.
Samples/groups are as previously explained. Different uppercase and lowercase letters indicate significance
of difference at $p < 0.01$ and $p < 0.05$, respectively. Same letters indicate no significant difference.

## Abundance and significance difference between the three groups at the genus level

The most abundant bacteria at the genus level were *Bacteroides*, with 15%, 13%, and
15% abundances in the CTR, SD, and LD groups, respectively. This was followed by
unclassified *Ruminococcaceae* at 13%, 14%, and 14% abundances in the CTR, SD, and
LD groups, respectively (Fig. 5). Other abundant genera included *Ruminococcus* (CTR:
14%, SD: 9%, LD: 10%), unclassified *Clostridiales* (CTR: 9%, SD: 11%, LD: 12%), and
*Faecalibacterium* (CTR: 8%, SD: 10%, LD: 8%). Ten genera were significantly ($p < 0.01$)
different in abundance between the CTR and SD groups, 7 between the CTR and LD
groups, and 5 between the SD and LD groups. Also importantly the genus *Aeriscardovia*
was significantly more abundant ($p < 0.01$) in the LD than in the SD and CTR groups
(Fig. 6). Interestingly, *Megamonas*, *Ochrobactrum*, and *Selenomonas* were significantly
more abundant ($p < 0.01$) in the CTR group than in the other two groups. *Aeriscardovia*,
*Delftia*, and *Rikenella* were significantly more abundant ($p < 0.01$) in the LD group than in
the CTR and SD groups. *Lactococcus* and *Fusobacterium* were significantly more abundant
($p < 0.01$) in the SD group than in the other two groups (Fig. 6). A heat map indicating
significantly expressed genera is shown in Fig. 7.

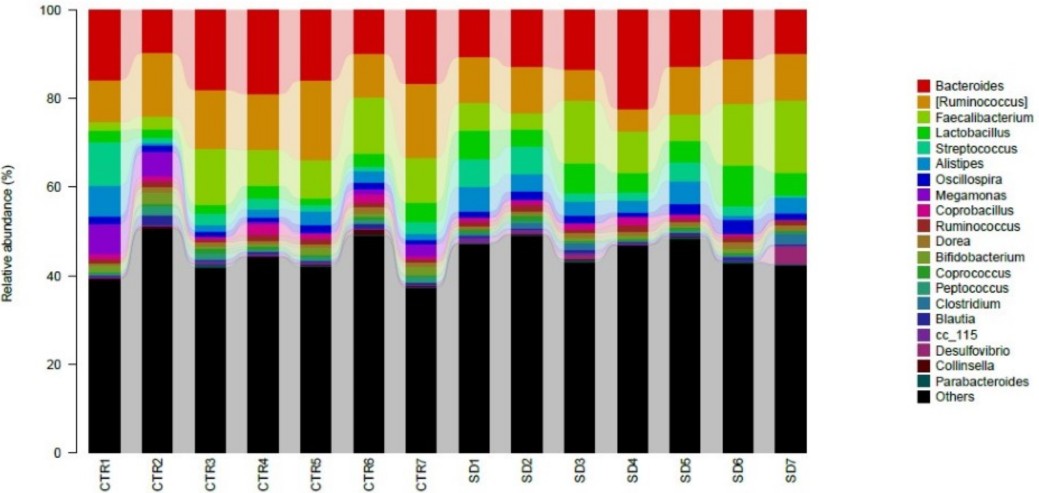

**Figure 5  Taxonomic profiles of the microbial communities at the genus level.** Samples/groups are as previously explained.

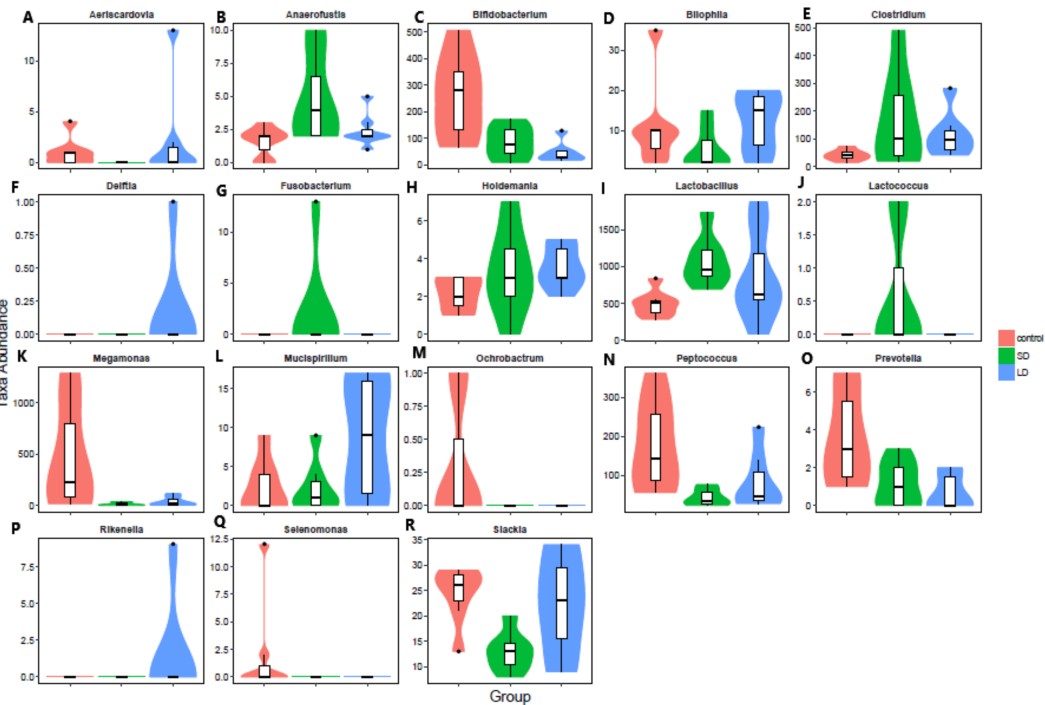

**Figure 6  Taxonomic profiles of the notable significant different bacterium at the genus level.**
(A) Aeriscardovia, (B) Anaerofustis, (C) Bifidobacterium, (D) Bilophila, (E) Clostridium, (F) Delftia, (G) Fusobacterium, (H) Holdemania, (I) Lactobacillus, (J) Lactococcus, (K) Megamonas, (L) Mucispirillum, (M) Ochrobactrum, (N) Peptococcus, (O) Prevotella, (P) Rikenella, (Q) Selenomonas, (R) Slackia. Samples/groups are as previously explained. Different uppercase and lowercase letters indicate significance of difference at $p < 0.01$ and $p < 0.05$, respectively. Same letters indicate no significant difference.

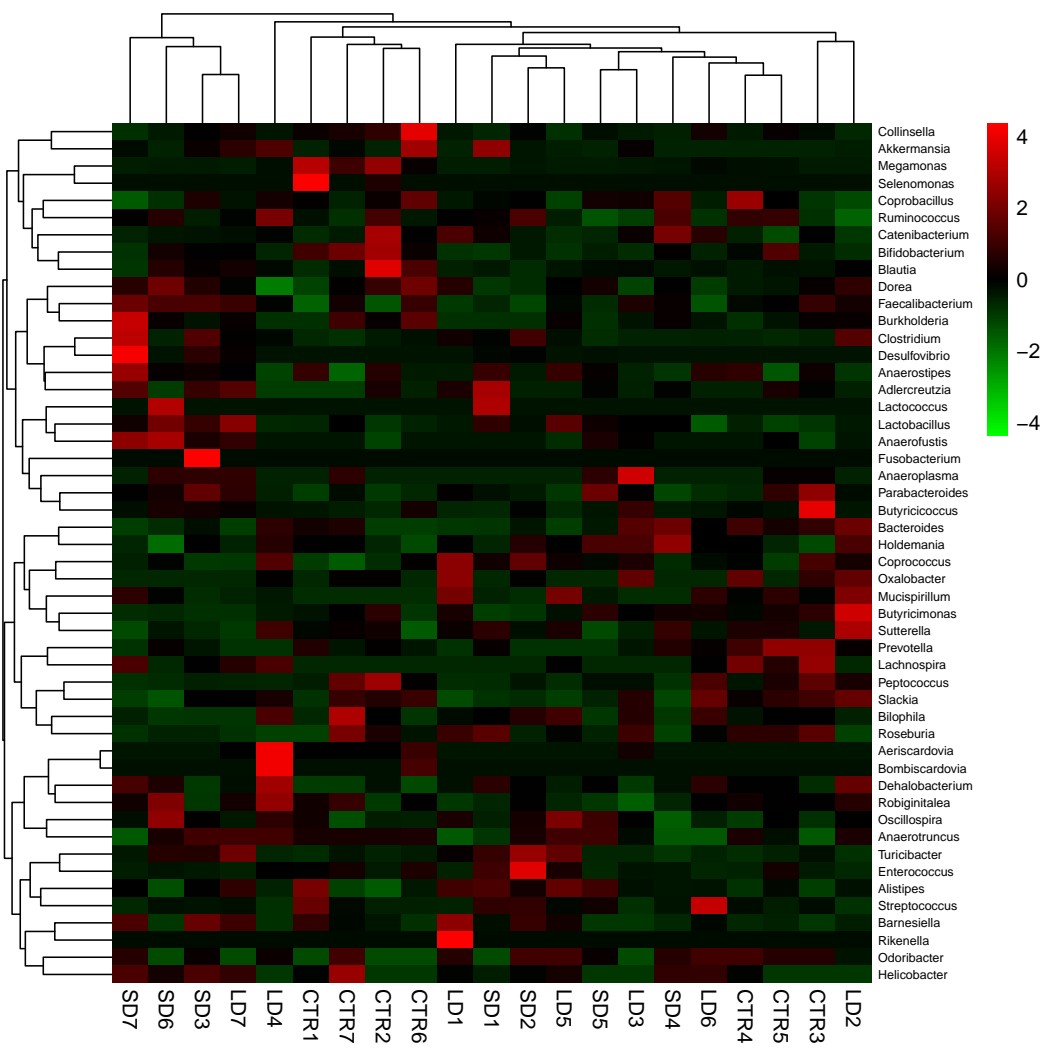

**Figure 7** **Heatmap showing the genera with significant differences of relative abundances amongst the three groups.** Heatmap is color-coded based on the scale of −4 to 4.

## Differences in predicted functional properties between the three groups

The differences in the effect of photoperiod on the functional properties across the three groups were further evaluated. Moderate differences were observed in cellular processes, particularly, in cell motility (Fig. 8). Cell motility was relatively low in samples from the CTR group compared to the motility in samples from the SD and LD groups. However, there were no notable differences in other functions such as transport and catabolism, cell growth and death, and cell communication across the groups. Analysis of the metabolism of the samples showed that carbohydrate metabolism was enhanced in CTR samples as compared to those in the SD and LD samples. Other functions did not exhibit any differences across the three tested groups (Fig. 9). Similar results were observed in other functional processes, such as genetic information processes and environmental information processes (results not shown).

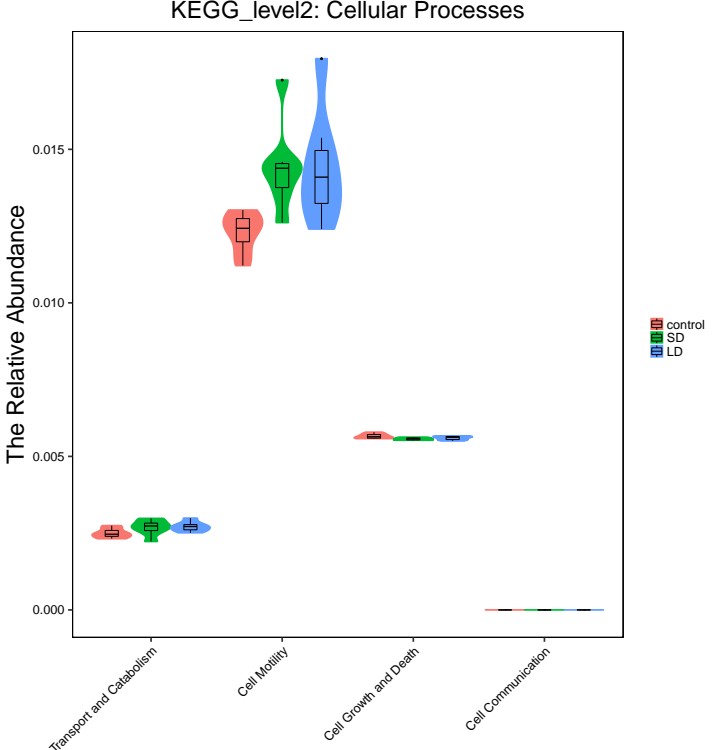

**Figure 8  Representing the functional differences at the cellular processes.** Samples/groups are as previously explained.

## DISCUSSION

Photoperiodism provides animals with the ability to change many physiological aspects and, consequently, adapting their body to the environment depending on the duration of light exposure (*Bailey et al., 2010*). The current study evaluated the structure and functional properties of the cecal microbiota of roosters that were subjected to three different photoperiodic regimes. Our data indicates that the length of time of light exposure may affect the abundance of specific bacteria in the cecum, leading to possible changes in functional properties. These changes may range across a variety of aspects the underlying mechanism of which has not been sufficiently explored.

To the best of our knowledge, there are few reports on the effects of photoperiod on gut microbiota in chickens and even in other species. Recently, a study demonstrated the role of photoperiod in changing gut microbiota. It indicated that different photoperiodic regimes (8 h dark/16 h light, 12 h dark/12 h light, and 16 h dark/8 h light cycles) could shape the gut microbiota of mice and thereby affect host radio sensitivity (*Cui et al., 2016*). The results of the present study are in general agreement with the findings of previous studies. Previously, it was reported that *Firmicutes* and *Bacteroidetes* dominate the broiler gut microbiota (*Cui et al., 2017*; *Oh et al., 2017*; *Zhou et al., 2017*), although effects of photoperiod on their abundances were not demonstrated. It was noted that *Megamonas* was significantly more

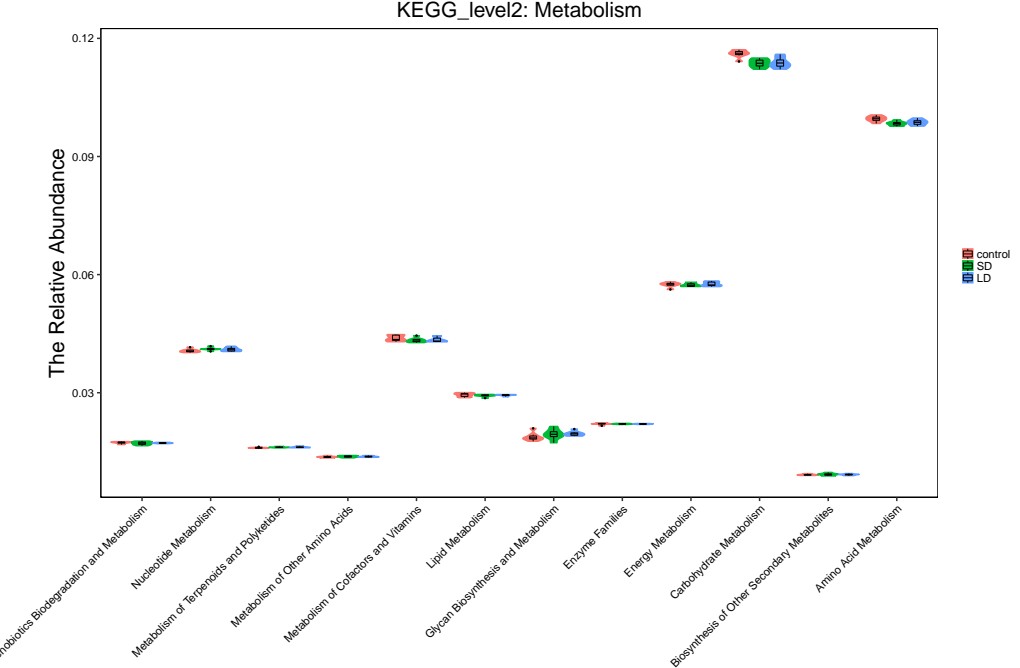

**Figure 9  Representing the functional differences at the metabolism level between the three groups.** Samples/groups are as previously explained.

abundant in CTR group as compared to LD and SD groups. It is of interest to note that *Megamonas* has been previously reported to play a significant role in fermenting glucose into acetate and propionate, which is pivotal for health benefits in a few species such as humans and ducks (*Chevrot et al., 2008*; *Sakon et al., 2008*; *Zhang et al., 2013*). These findings may indicate a new platform for manipulating acetate and propionate in broiler roosters by varying photoperiod regimes. Short term (LD) exposure to light has also been indicated to significantly increase the abundance of the novel genus *Aeriscardovia* (*Simpson et al., 2004*) and of the gram-negative bacteria *Delftia*, which has been reported to be associated with infectious diseases (*Bilgin et al., 2015*; *Calzada et al., 2015*). Long term (SD) exposure to light significantly increases the abundance of gram-positive *Lactococcus*, which has been reported to have potential for use in preventing infectious diseases (*Hanniffy et al., 2007*). The gram-negative genus *Fusobacterium*, which has been reported to be associated with infections in humans (*Kostic et al., 2013*), was also significantly more abundant under an SD regime. Our results seem to indicate that short term (LD) photoperiod (8 h light) could increase the abundance of bacterial genera associated with infectious diseases in the rooster gut, while long term photoperiod (16 h light) could increase the abundance of bacterial genera associated with preventing infectious diseases. However, this deduction needs to be verified by more extensive scientific investigation. These results are of importance to prompt more studies on the role that photoperiod may play with regards to physiology in animals. It is of note that different results may arise due to differences in time of exposure, light intensity, animal and growth stages, and other factors employed in the study.

The effect on several functional properties of the roosters may be attributed to increases in the abundance of specific bacteria caused by light exposure duration. Analysis of predicted functional properties in the present study indicated that metabolism may be influenced by photoperiod. Carbohydrate metabolism was enhanced in the CTR group, as compared to the SD and LD groups. Previous studies have demonstrated that gut microbiota plays an important role in the life activities of chickens (*Waite & Taylor, 2014*). The present study is limited by the fact that the change in gut microbiota was not correlated with performance (e.g., testis development or body weight). Further studies are suggested to investigate the effects of photoperiod on gut microbiota and their relationship with growth or reproduction performance.

## CONCLUSIONS

Our results indicate that photoperiod may affect the abundance of specific bacteria in the gut and thereby contribute to differences in the functional properties of the gut microbiota in broiler roosters.

### Funding

This work was supported by National Key Research and Development Program of China (2016YFD0500502). The funders had no role in study design, data collection and analysis, decision to publish, or preparation of the manuscript.

### Grant Disclosures

The following grant information was disclosed by the authors:
National Key Research and Development Program of China: 2016YFD0500502.

### Competing Interests

The authors declare there are no competing interests.

### Author Contributions

- Jun Wang and Lucky T. Nesengani conceived and designed the experiments, analyzed the data, wrote the paper, prepared figures and/or tables.
- Yongsheng Gong performed the experiments, contributed reagents/materials/analysis tools.
- Yujiang Yang contributed reagents/materials/analysis tools, reviewed drafts of the paper.
- Wenfa Lu conceived and designed the experiments, reviewed drafts of the paper.

### Field Study Permissions

The following information was supplied relating to field study approvals (i.e., approving body and any reference numbers):

All experiments were approved by the Jilin Agricultural University Ethical Committee.

## Data Availability

Lu, Wenfa (2018): seqs.fna.gz. figshare. http://dx.doi.org/10.6084/m9.figshare.5848077.v1.

## Supplemental Information

Supplemental information for this article can be found online at http://dx.doi.org/10.7717/peerj.4390#supplemental-information.

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
