# Peer review of "S rRNA gene sequencing reveals effects of photoperiod on cecal microbiota of broiler roosters"

_PeerJ, doi:10.7717/peerj.4390_

## Round 0.1 · original submission · Major Revisions

I believe both reviewers have raised important issues related with poor use of language, missing details in the introduction, and over interpretation of the data. They provide general and specific comments that must be addressed during revision.

Reviewer 1 ·

Basic reporting

The study conducted by Wang et al. describes the impact of three different light regimens on the cecal microbiota profiles of Broiler Roosters. Overall, the topic is very interesting. However, the way the microbiota data has been analyzed, presented, and interpreted, as well as the shallow depth of the introduction and discussion sections, raise many concerns. English language also needs to be improved. Please find below my specific comments.

Abstract:

- Abstract should be self-explanatory. Define the abbreviations for the treatment groups (LD, CTR, SD, what do they stand for?)


Introduction:

- line 48: Consider paraphrasing " most of pivotal roles". You mean different aspects of the physiology?

- lines 55-56: photoperiod alters with the community structure of gut microbiota???

- line 57: Gut microbiota is not only made up of bacterial species, but different groups of archaea, fungi, viruses,,,,

- line 63: delete "the role of".

- line 64: most functions of what?

- Line 65: knowledge is limited? So, do you mean there is any other manuscript published in this field? If yes, it should be mentioned in the introduction. If not, then it is a knowledge gap, not limitation.

- Lines 65-67: not common to say "the role of something in the abundance of something else"

- line 66: this study is not evaluating the functional differences. It should be clearly mentioned that "predicted functional properties or genes" have been evaluated. Acknowledge the limitation of prediction in first place. Second, the authors should keep in mind that even if they have done actual metagenomics (shotgun sequencing), changes in functional properties (genes) do not necessarily translate to changes in the functions of microbiota. Interpretation of PICRUSt data should be practiced very carefully in order to avoid provision of misleading information to the general audience of this manuscript that are not familiar with microbiome science.

Overall, the introduction is poorly written, scientific-wise and language. There is no clear link between different sections of the introduction. The authors should identify potential mechanisms by which photoperiod may modulate the composition of gut microbiota. Doing a brief search suggests that in other species (e.g. mice) hormonal changes and gut-brain-axis can play role. The most important link, particularly in broiler, would be the photoperiod-mediated shifts in the feed intake. Of course diet and availability of nutrient are the most important contributing factors to the overall structure of gut microbiota. There is a wealth of knowledge available regarding the influence of photoperiod on the production/performance parameters of broilers (feed intake, FCR, weight gain, etc.) that is being ignored throughout the introduction, and how they can be associated with shifts in the composition of microbiota.

Experimental design

Materials and Methods:

- Lines 69-70: there should be a protocol number or precise reference number for this approval.

- Lines 74-78: Why "LD" is used for Short Day and "SD" for Long Day???

- Lines 79-81: How many broilers in each cage? Does this mean that the feed intake for all broilers has been equal throughout experiment? This needs to be clarified in order to help interpreting the data. If feed intake and feed composition has been equal for all, then compositional shifts in the microbiota can be linked with other factors (e.g. hormonal, gut-brain-axis, etc.).

-Line 80: what is the composition of this "commercial diet"? It should be provided as a table (at least supplementary file). This will be a required infromation for future comparisons of the result of this study with others.

- As a general note, why performance parameters are not accompanying the microbiota data? Weight gain and FCR should have been measured for these broilers. This can improve the analysis/presentation of the microbiota data as well (correlation analyses between performance parameters and microbiota composition/predicted functions).

- line 86: " refrigerator" change to freezer.

- line 96: change 35 to 25

- Lines 111 & 121: Sequence analysis is done using Bioinformatics. Methods should be presented as one section for "Sequence analysis and Bioinformatics" and another section for " Statistical analyses". Or you can combine both.

- Line 126: weighted or unwieghted UniFrac?

- Line 133: "Predicted functional genes" instead of "microbial functions"

- Lines 111-120: What OTU picking strategy was used? Reference-based or De-novo? If reference-based (closed-reference) script of QIIME, then what percentage of reads were not mapped to database and discarded? This is the main criteria to be considered for validity of PICRUSt data.

- Why not using PCoA or NMDS for visualizing the beta-diversity of microbiota?

Validity of the findings

Results:
-Line 138: Why about?

-Lines 139-141: Not informative. Instead of total and per group reads, the sequencing depth should be specified based on "average number of reads per sample within each treatment group".

-Lines 143-145: these are material methods rather than results.

-Lines 146-147: What is the benefit of reporting per sample data without performing among groups comparisons???

-Line 151: bacterium? change to bacterial phyla.

- Line 161: the other abundant genera,,,

- Line 163: 10 genera, line 165: 5 genera , fix this throughout the text.

-Line 174: why heatmap of genera is reported here in the PICRUSt result?

_line 172: again, these are not functional differences. This should be fixed throughout the manuscript, specifically the discussion.

Discussion:

Discussion in general is very descriptive, repeating the results in most parts. Lots of speculations and non-relevant comparisons between different species of animals.

-Lines 204-207: this is a very loose speculation, changes in the proportion of one genus cannot be translated to change in the metabolite profile of the gut.

- Line 208: Indicate that you are comparing this with swine studies, etc. And, what is the relevance?

-Line 212: Lactococcus as a metabolite??? There is for sure better way for describing the beneficial role that lactic acid bacteria can play in gut health.

Reviewer 2 ·

Basic reporting

The study investigated impact of three different photoperiod regimes on cecal gut microbiota in rooster broiler chickens. It is admirable that the authors investigated an important aspect that has not been greatly explored in the past. However, that said, I have several major and minor concerns with details of the manuscript, and with the author’s interpretations. Some of these concerns are highlighted here and in the sections that follow.

The English language is a major concern I have with this manuscript, particularly in the introduction section, discussion and in other sections as well (some specific areas are indicated below). This has a major impact on the flow of the text and needs to be improved. For example, the authors can utilize the services of English editors.

The introduction does not clearly and adequately capture the existing literature and clearly state the knowledge gap in the topic of the study. English language is also a major weakness in this section and should be improved. This to some extend applies to the discussion section, which should be improvised as it is not well detailed in relating the current results with literature.

The structure of the manuscript is adequate and the figures are of high quality; however, table 1 should be revised to report statistically analyzed data.

Experimental design

Presentation of the study design is okay but it could have been improved by providing detailed descriptions on various aspects of the study. Specific areas that needed to be improved are shown in general comments section. However, the research question is clearly stated and the study addresses an important area of research.


The authors collected luminal contents from the cecum only. Could they explain why mucosal samples from the same section were not collected/analyzed? Interpreting temporal changes found in the luminal contents, in the absence of mucosal changes, especially when the birds were indeed slaughtered can be weak without a justification.

The methodologies and statistical analysis used in this study are commendable. However, there are several areas that needs to be improved upon, regarding reporting and presentation of the information (see my comments below). I find some of the description of the analysis methods/ techniques used inadequate and or unclear

Validity of the findings

My concern is the over-interpretation of the data in two primary regards. One, there is no evidence whether the changes observed at phylum level, and at genus levels are indeed due to the light regimes because these could be temporal changes related to diet, housing or generally the management, as there was no baseline analysis that were done in this study to prove that the birds had a similar microbiota profile before the start of the experimental procedures. Second, the interpretation of the data at functional level is to some extent overstated given that very minor changes were observed at this level.


Part of the literature cited in the discussion is not clear how it relates to the current results and may not apply to the current study (e.g Line 213 Fusobacterium in humans). Line 219 – 220, the rainbow trout study doesn’t mention changes in gut microbiota. However, the authors have acknowledged that these kind of speculations needs further investigations

The abstract conclusion is quite broad and doesn’t clearly capture the main message or conclusion of the study, given its scope. However, this is somehow corrected in the overall conclusion of the discussion section. I suggest that the abstract conclusion be revised to be in line with the overall conclusion of the results reported in this study.

Additional comments

Line 32: Even though the descriptions of the different light regimes used in this study are clearly indicated, the authors did not provide the full meaning of the abbreviations used, as this is the first time they appear in the paper (LD, CTR, SD)

Line 42 – 43: The way this sentence is written is confusing and not clear. It can be improved by revising the sentence structure/grammar


Line 43 – 44: Based on the scope of this study, the abstract conclusion is quite broad and doesn’t clearly capture the main message or conclusion of the study

Line 50: It is not clear what message the authors are trying to put across here. Revising the English language here can improve the sentence

Line 55 – 56: Photoperiod may be,,,,,,,(modify, move or delete this sentence). As it is, it doesn’t have a good connection with the previous or succeeding sections and hence, it interferes with the flow

Line 57 – 65: This section is very confusing and difficult to understand. There is need to improve on the English language (sentence structure, choice of words, flow, etc), as well as on the message the authors are trying to put across.


Line 74 – 76: The abbreviations for the different light regimes are confusing (long day photoperiod; SD, and Short day photoperiod; LD. Why not the other way?

Group II: Long day photoperiod (SD; 16 h Light: 8 h Dark, i.e. lights on at 04:00 p.m. and lights off at 08:00 a.m). why were the lights not switched on at 8: 00 am as in the other two groups? Also, please explain how was darkness maintained in this group given the fact that the lights were off during the day time?

Line 79: Were the birds housed in individual cages or group? If in group cages, how many birds per cage? Was the 115g/day per bird (in case of individual housing) or per cage (in case of group)

In line 73, the authors mentioned that the birds were kept for 5 weeks, but in line 80, they talk of 20 weeks. Was 20 weeks the period prior to the experiment? Please clarify this. Also, clarify the fact that the feed intake was increased by 5 g every week. Did the increase start after 20 weeks of age (during the experimental period) or when exactly did it start and until when?

Line 83: The authors here mention that the roosters were slaughtered at about 24 weeks of age, which kind of contradicts lines 72 – 73 where they mention 20 weeks of age and 5 weeks photoperiod regime. Please clarify

Line 87: 16S rDNA is mentioned here but everywhere else 16S rRNA gene is used. Try to be consisted and correct this to 16S rRNA

Line 93: gene, not genes

Line 107: The authors mention that Illumina Miseq platform was used for sequencing; however, earlier they indicated that pyrosequencing was used (line 67). Please clarify

Line 122 – 127: Here and in other sections of the manuscripts, the word ‘the’ is overused

Line 138 – 141: Also, include the average number of sequences per sample

Line 142: the subtitle (Validation and structure of the sequences) doesn’t agree with the message communicated in lines 143 – 147. Here the authors mention diversity analysis, which explains microbial community composition/structure as well as diversity indicators such as richness and evenness, which could not be necessarily a validation or structure of sequences.

Also, the authors mention significant differences in data distribution (P<0.01) under unweighted Unifrac distance, but there are no figures/data in support of this e.g PCoA plots that could be associated with the ANOSIM P-value provided here.
It is also not mentioned whether the data is shown or not, because Table 1, which is quoted here shows data for alpha diversity but not Beta diversity.

It is possible that the significance difference reported here could be due to the housing/cage effect, and without the data and or clear statistical analysis reporting it is difficult to attach any significant impact to this kind of results.

The authors need to be clear on the diversity measures that they used. As it is now, they don’t seem to understand the distinction between different diversity measures (e.g beta diversity and alpha diversity)

Table 1: The title should be improved to clearly state the message in the table. Also, the table only shows raw data of the different diversity indices used for each sample, but does not indicate any statistical analysis.

Line 148: at the phylum level
Line 149 and 159: the word occupying is inappropriate
Line 158: at the genus level

Line 173: at functional level

Line 173 – 174: “A heat map indicating significantly expressed genus is shown in Fig.5”. This section is reporting functional level differences but not changes at genus level. Move this sentence to the previous section

Line 202: ,,,some bacteria in rooster gut. As such,

Line 206 – 209: Revise grammar in this section and restructure the sentences

---

## Round 0.2 · Minor Revisions

Please address the few remaining issues raised by reviewer #1. Once those are addressed, the manuscript would be a lot clearer.

Reviewer 1 ·

Basic reporting

No comment

Experimental design

No comment

Validity of the findings

No comment

Additional comments

Figure 1 and 2, legends: "Samples are presented along with the horizontal axis and relative abundance at the vertical axis" this description is totally wrong. Please refer to literature to find out more about the definition of these axes and how to describe nMDS and PLS-DA analyses.

Line 180: the most abundant ,,,

Lines 163-165: "The alpha-diversity indices (chao1, Simpson and Shannon index) are reported in Table 1. The results of the beta diversity analysis and the PLS-discriminant analysis are shown in Fig. 1and 2, respectively." You need to report important findings of these table/figures. The table itself is not associated with any descriptive caption. It also does not statistically compare the indices of alpha-diversity among groups.


Lines 224-227: " With the exception of Firmicutes and Bacteroidetes, our results suggest that different photoperiodic regimes could dictate the abundance of some bacteria in broiler rooster gut microbiota. It was noted that Megamonas was significantly more abundant in CTR group as compared to LD and SD groups." Isn't Megamonas a Firmicutes genus? Then why "with the exception of Firmicutes" ? Many of bacteria that you have reported to shift in response to photoperiod belong to Firmicutes and Bacteroidetes.

Lines 253-257: " These findings provide a baseline to be used in further investigation of the factors that may be detrimental in the regulation and control of the functional properties of the gut microbiota. To the best of our knowledge, this is the first report on the effects of photoperiod on chicken gut microbiota which may shed some light on future studies in the field. " these sentences do not add value to the discussion, too vague.

Reviewer 2 ·

Basic reporting

The authors addressed all comments adequately

Experimental design

Required details and clarifications were provided

Validity of the findings

Missing information was provided and all concerns addressed

Additional comments

The authors did a good job in revising the manuscript

---

## Round 0.3 · accepted · Accept

I appreciate all the revisions you made.